# META-LEARNING FOR SUBJECT ADAPTATION IN LOW-DATA ENVIRONMENTS FOR EEG-BASED MOTOR IMAGERY BRAIN-COMPUTER INTERFACES

**Arnav Pati**[*]  **Deepak Mewada**[*]  **Dr. Debasis Samanta**[*]

## ABSTRACT

Motor imagery classification from Electroencephalogram (EEG) signals involves decoding information during the imagination of specific movements. However, learning representations for EEG-based motor imagery classification is challenging due to inter-subject variability and differences in mental imagery, resulting in poor generalization of deep learning models to new subjects. While pre-trained deep learning models achieve high accuracy on subjects with similar domains, they fail on subjects with dissimilar domains. Optimization-based meta-learning algorithms can address this limitation by learning a good initialization for the model, enabling quick adaptation to new subjects with limited fine-tuning examples. We demonstrate that our Meta Learning approach consistently outperforms Transfer Learning on the BCI Competition IV 2a dataset. Although accuracy varies depending on domain similarity, meta-learning demonstrates efficient adaption to unseen subjects with limited data. By improving generalization across subjects with different domains under low-data environments, we can enhance the reliability and practicality of brain-computer interfaces for real-world applications.

## 1 INTRODUCTION

Deep learning models have shown great success in many fields, including image and speech recognition. However, applying these models to new domains or tasks can be challenging, as they often require large amounts of labeled data for training. One approach to address this challenge is to use transfer learning, where a model pre-trained on a large dataset is fine-tuned on a smaller, task-specific dataset. Learning representations for EEG-based motor imagery classification (Guragai et al. (2020)) is a challenging task due to the differences in mental imagery and inter-subject variability (Saha Simanto (2020)). In this paper, we investigate the effectiveness of meta-learning in improving generalization to new subjects with limited data, in the context of EEG-based motor imagery classification. We compare the performance of our Meta-Learning approach with basic transfer learning on the BCI Competition IV 2a dataset and show that our approach outperforms the baseline on subjects with dissimilar domains.

Previous studies (Zhang (2020)) have used transfer learning to address the problem of inter-subject variability in EEG-based motor imagery classification. However, these approaches are limited by the similarity of the domains between the training and testing subjects. Optimization-based meta-learning algorithms (Hospedales et al. (2022)) like Reptile (Nichol et al. (2018)) can be used to learn a good initialization for the model to adapt quickly to new subjects with few fine-tuning examples, which can improve the learning of representations. In recent research, several attempts have been made to address similar challenges using meta-learning-based approaches. For instance, Li et al. (2021) demonstrated the superiority of meta-learning; however, their work employed Model-Agnostic Meta-Learning and did not conduct subject-specific or shot-wise analysis. In another study, Wu & Chan (2022) utilized a Reptile-based approach for motor imagery (MI) classification but reported its ineffectiveness without providing detailed explanations for the observed outcomes. Our contribution lies in demonstrating the superiority of our meta-learning approach compared to basic transfer learning. Through subject-specific and shot-wise analyses, we effectively address subject variability with limited data. By applying meta-learning to EEG-based motor imagery classification, our work contributes to the expanding research on improving generalization across subjects in diverse domains, even with limited data availability.

## 2 METHODOLOGY

To evaluate our approach, we focus on the ability to generalize to an unseen subject. We use the data from all other subjects in the dataset as our training dataset, while the data from the unseen test subject is divided into fine-tuning and testing datasets.

---

[*]All authors are from the Department of Computer Science and Engineering, Indian Institute of Technology, Kharagpur. For correspondence contact deepakmewada96@kgpian.iitkgp.ac.in

Our approach involves three stages. First, we train the model on the training dataset using a pre-training algorithm (Transfer Learning / Meta Learning). Then, we fine-tune the model using gradient descent on a few shots of trials from the fine-tuning dataset. Finally, we test the performance of the model on 4 shots of trials from the testing dataset. We utilize the EEGNet (Lawhern et al. (2018)) architecture as the backbone of our approach. During few-shot fine-tuning on the test subject, we freeze the feature extraction layers and only fine-tune the classifier.

Our Meta-Learning algorithm is employed for meta-training on the 8 training subjects to update the model's parameters ($\theta$). In each iteration, we sample 5 mini-batches of tasks, each from a different subject, consisting of a support set $S$ and a query set $Q$. Separate model copies are created for each mini-batch, and their parameters ($\theta'$) are updated through 2 steps of gradient descent on $S$. The loss is then evaluated on the updated parameters ($\theta'$) using $Q$, and the gradient of this loss is calculated independently for each mini-batch. The sum of these gradients is used to update the main parameters $\theta$. For a more detailed algorithm, please refer to Algorithm 1 in the Appendix. Through this approach, our goal is to learn an effective initialization for the model, enabling rapid adaptation to new subjects with limited fine-tuning examples.

## 3 EXPERIMENTS

We used the BCI COMPETITION IV 2a Dataset (Brunner et al. (2008)), consisting of EEG signals from 9 subjects (numbered 1 to 9). The dataset comprises 4 motor imagery classes (left hand, right hand, feet, tongue), with 144 trials per class for each subject. Each trial has a duration of 4 seconds, sampled at 250 Hz, resulting in 1001 timepoints. The EEG data is recorded using 22 channels. To evaluate the generalization of our approach, we performed 9 separate experiments, where we pre-trained on the EEG signals from 8 subjects and tested on the remaining subject. Specifically, for each $i = 1$ to 9, we used subject $i$ as the test subject and pre-trained on the EEG signals from the other 8 subjects. We compared the performance of our Meta Learning approach with a transfer learning approach that utilizes gradient descent on EEGNet for pre-training. To evaluate the accuracy achieved by our approach on subject $i$, we compared its performance using a few shots of fine-tuning examples against the performance achieved when trained on a larger number of fine-tuning examples. For this purpose, we divided all trials of subject $i$ into two sets: a fine-tuning set and a testing set. Subsequently, we fine-tuned our pre-trained model, while freezing the feature extraction layers, using the few shots of trials from the fine-tuning set and evaluated the accuracy on the testing set. In order to assess many-shot classification accuracy, we divide the test subject's data into 80% for fine-tuning (112 trials) and 20% for testing. Through training the pre-trained model on additional data from the new subject, we simultaneously fine-tune the feature extractor layers. Our primary aim is to evaluate the effectiveness of our Meta Learning approach in achieving comparable accuracy to the many-shot accuracy, despite utilizing only a few fine-tuning trials.

## 4 RESULTS AND CONCLUSION

Table 1: Average Accuracy (in %) of Transfer Learning and Meta Learning Approaches

| Method↓ Shots→ | 0 | 2 | 4 | 6 | 8 | 10 | many (112) |
|---|---|---|---|---|---|---|---|
| **Transfer Learning** | 49.67 | 51.85 | 54.44 | 56.00 | 56.77 | 59.59 | 79.66 |
| **Meta Learning** | **55.56** | **55.38** | **57.52** | **59.67** | **60.94** | **63.13** | **82.24** |

Table 1 shows the average accuracy across each of the 9 experiments. Our Meta Learning approach consistently Transfer Learning in terms of few-shot classification accuracy on the BCI Competition IV 2a dataset, indicating its ability to learn a better initialization of EEGNet parameters and effectively learn representations using limited fine-tuning data. We observe that accuracy increases as the number of fine-tuning shots increases, but the few-shot accuracy remains significantly lower than the many-shot accuracy, highlighting the need for extensive fine-tuning on unseen subjects. Meta Learning generally outperforms Transfer Learning even in many-shot classification scenarios. However, our observations reveal that the accuracy of many-shot classification can vary depending on the similarity between test subjects and training subjects, including in the case of Meta Learning. For a more detailed analysis on various subjects, please refer to the Appendix.

In conclusion, our Meta Learning approach successfully addresses the challenges of subject variability and demonstrates its effectiveness in learning better representations on adaption to unseen subjects with limited fine-tuning data. Our work opens up new possibilities for enhancing the reliability and efficiency of communication with external devices through BCI technology.

URM STATEMENT

Authors Arnav Pati and Deepak Mewada meet the URM criteria of the ICLR 2023 Tiny Papers Track.

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

# A  APPENDIX

**Code and Implementation Details:** The code and implementation details can be accessed at the following GitHub repository: `https://github.com/arnav-pati/iclr2023-meta-learning-eeg-mi-classification`.

---

**Algorithm 1** Meta-training

---

**Require:** Meta-training dataset $D$, EEGNet model $f_\theta$ with parameters $\theta$, Loss function $L(\cdot)$, hyperparameters $\alpha, \beta$, Number of inner-loop gradient update steps $K$
**Ensure:** $\theta$: Updated model parameters
1: **for** iteration in iterations **do**
2:     Sample batch $B$ from $D$
3:     **for** subject in $B$ **do**
4:         Sample support set $S$ and query set $Q$ from subject
5:         Initialize $\theta' \leftarrow \theta$
6:         **for** $k$ in $1, 2, \ldots, K$ **do**
7:             Compute embeddings for $S$ and $Q$ using $f_{\theta'}$
8:             Update model parameters $\theta'$ using gradient descent with $\nabla_{\theta'} L(S; \theta')$
9:         **end for**
10:       Compute loss $L_{\text{subject}}$ using $L(Q; \theta')$
11:     **end for**
12:     Update model parameters $\theta$ using gradient descent with $\displaystyle\sum_{\text{for } subject \text{ in } B} \nabla_\theta L_{\text{subject}}$
13: **end for**
14: **return** $\theta$

---

Table 2: Performance Analysis of Transfer Learning and Meta Learning Approaches for Classification Accuracy (in %) across Different Fine-tuning Shots and Test Subjects

| | Fine-tuning shots | | | | | | | | | | | |
|---|---|---|---|---|---|---|---|---|---|---|---|---|
| | 0 | 1 | 2 | 3 | 4 | 5 | 6 | 7 | 8 | 9 | 10 | many(112) |
| **Test Subject 1:** | | | | | | | | | | | | |
| Transfer Learning | 63.54 | 60.71 | 64.32 | 66.88 | 68.75 | 69.53 | 69.20 | 70.19 | 71.35 | 73.86 | 72.50 | 83.93 |
| Meta Learning | **70.83** | **64.96** | **66.15** | **69.06** | **70.49** | **71.09** | **73.21** | **74.52** | **73.96** | **75.00** | **76.25** | **90.18** |
| **Test Subject 2:** | | | | | | | | | | | | |
| Transfer Learning | 32.81 | 36.16 | 34.11 | 34.06 | 36.46 | 38.28 | 38.84 | 41.83 | 39.58 | 39.77 | 41.88 | 67.86 |
| Meta Learning | **35.42** | **36.38** | **39.58** | **38.44** | **41.67** | **41.41** | **41.96** | **42.79** | **43.75** | **43.75** | **46.25** | **73.21** |
| **Test Subject 3:** | | | | | | | | | | | | |
| Transfer Learning | 65.10 | 65.40 | 67.97 | 70.31 | 70.49 | 71.88 | 71.88 | 73.08 | 73.44 | 72.73 | 77.50 | 91.96 |
| Meta Learning | **71.88** | **70.54** | **72.92** | **73.75** | **76.39** | **79.30** | **79.02** | **80.29** | **81.25** | **78.98** | **81.25** | **97.32** |
| **Test Subject 4:** | | | | | | | | | | | | |
| Transfer Learning | 41.15 | 43.97 | 46.35 | 47.50 | 49.65 | 48.83 | 53.13 | 52.40 | 51.56 | 52.84 | **59.38** | 82.14 |
| Meta Learning | **48.44** | **48.66** | **48.18** | **50.31** | **52.43** | **53.13** | **54.02** | **54.33** | **56.77** | **55.68** | 56.25 | **83.93** |
| **Test Subject 5:** | | | | | | | | | | | | |
| Transfer Learning | 34.38 | 32.59 | 34.38 | 34.69 | 36.81 | 38.28 | 37.05 | 37.50 | 35.94 | 39.20 | 38.75 | 56.25 |
| Meta Learning | **36.63** | **39.06** | **39.58** | **38.44** | **38.54** | **38.28** | **40.63** | **41.83** | **42.19** | **43.75** | **44.38** | **59.82** |
| **Test Subject 6:** | | | | | | | | | | | | |
| Transfer Learning | 42.71 | 39.51 | 43.49 | 43.44 | **45.83** | 44.92 | 46.88 | 44.71 | 46.88 | 46.59 | 51.88 | 66.96 |
| Meta Learning | **47.74** | **43.53** | **45.83** | **45.94** | 43.75 | **46.09** | **48.21** | **49.52** | **47.92** | **51.70** | **55.63** | **74.10** |
| **Test Subject 7:** | | | | | | | | | | | | |
| Transfer Learning | 50.52 | 47.54 | 51.04 | 53.75 | 53.47 | 55.08 | 54.91 | 55.29 | 55.21 | 57.39 | 60.00 | **91.07** |
| Meta Learning | **59.55** | **53.57** | **55.99** | **56.88** | **58.68** | **62.89** | **61.61** | **65.38** | **60.42** | **61.36** | **61.25** | 90.18 |
| **Test Subject 8:** | | | | | | | | | | | | |
| Transfer Learning | 57.99 | 58.93 | 61.46 | 62.81 | 63.19 | 64.84 | 65.18 | 65.38 | 67.19 | 69.32 | 66.25 | **89.29** |
| Meta Learning | **66.49** | **61.83** | **64.58** | **65.63** | **69.10** | **71.48** | **69.64** | **69.71** | **70.83** | **71.59** | **73.75** | 88.39 |
| **Test Subject 9:** | | | | | | | | | | | | |
| Transfer Learning | 58.85 | 58.26 | 63.54 | 63.44 | 65.28 | **68.75** | 66.96 | 69.71 | 69.79 | 66.48 | 68.13 | **87.50** |
| Meta Learning | **63.02** | **66.07** | **65.63** | **66.88** | **66.67** | 68.36 | **68.75** | **69.71** | **71.35** | **69.88** | **73.13** | 83.04 |

Table 2 presents an accuracy comparison for different numbers of fine-tuning shots across various test subjects using Transfer Learning and Meta Learning methods. Here are a few observations:

1. Generally, we observe that as the number of fine-tuning shots increases, the accuracy of both the Transfer Learning and Meta Learning models tends to improve across most test subjects.

2. Test Subjects 2, 5 and 6 exhibit low 0-shot accuracy with both Transfer Learning and Meta Learning, indicating their dissimilarity to the training subjects. However, Meta Learning demonstrates superior performance over Transfer Learning in fine-tuning with few shots and continues to outperform even after fine-tuning with a larger number of shots.

3. Meta Learning significantly outperforms Transfer Learning in learning representations from very few shots for Test Subjects 4, 7, 8, and 9. However, Transfer Learning gradually improves with more data, reaching comparable accuracy to Meta Learning in many-shot classification.

4. For Test Subjects 1 and 3, Meta Learning significantly outperforms Transfer Learning in learning representations from both few shots and many shots. This highlights that Meta Learning provides pre-trained parameters that consistently outperform Transfer Learning, even with a substantial amount of fine-tuning data.

Overall, the Meta Learning tends to outperform Transfer Learning across multiple test subjects and fine-tuning shot numbers, indicating its effectiveness in the given context. However, it is worth noting that the results may vary depending on the specific dataset and problem domain.

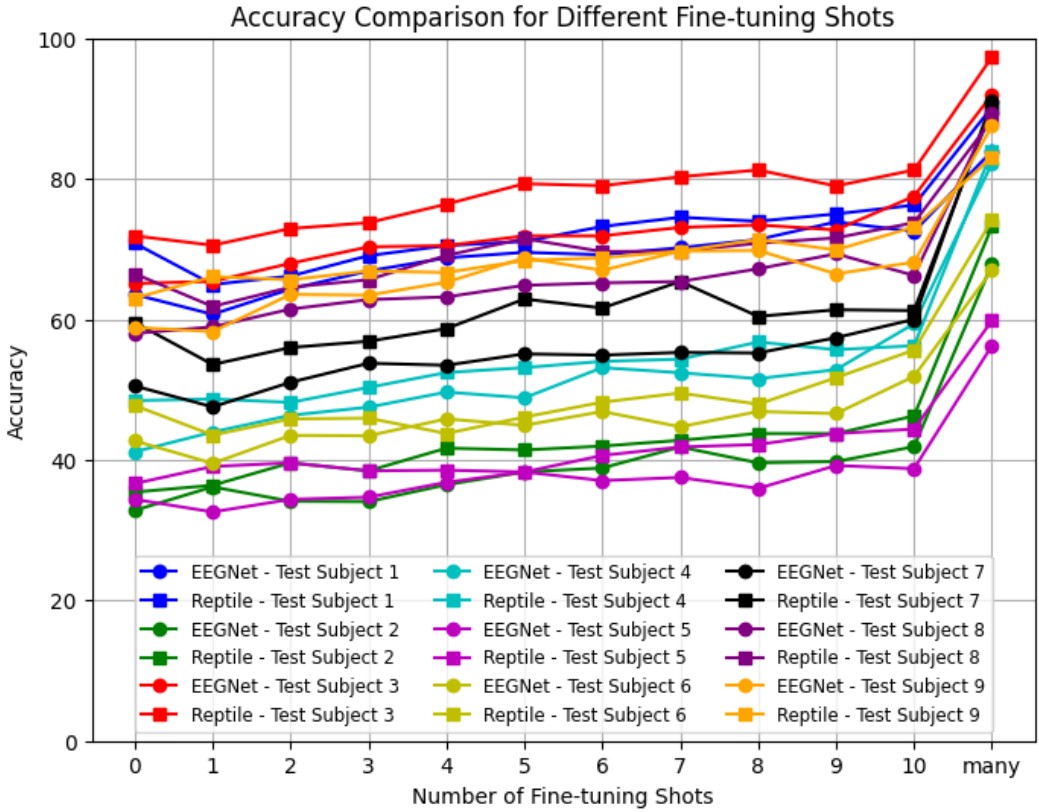

Figure 1: Performance Analysis by Shots and Comparison between

The figure1 compares the accuracy of Transfer Learning and Meta Learning, for different numbers of fine-tuning shots. The x-axis represents the number of shots, and the y-axis represents the accuracy. Each line plot represents a different test subject. The plot shows that, in general, increasing the number of fine-tuning shots improves the accuracy for both models. Transfer Learning typically requires a larger amount of data to achieve comparable accuracy to Meta Learning across multiple test subjects. However, there are instances where both models exhibit similar performance.

Figure 2 shows that the validation loss after fine-tuning with many shots is significantly lower when pre-trained using our Meta Learning approach.

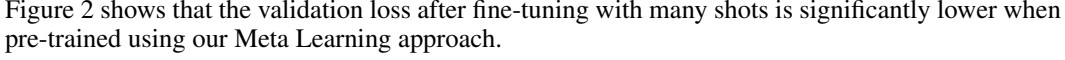

(a) Test Subject 1 (b) Test Subject 2 (c) Test Subject 3

(d) Test Subject 4 (e) Test Subject 5 (f) Test Subject 6

(g) Test Subject 7 (h) Test Subject 8 (i) Test Subject 9

Figure 2: Training the pre-trained model on a large amount of fine-tuning data from the unseen subject: The x-axis shows the number of epochs, and the y-axis shows the classification accuracy on testing trials.

