# OpenReview forum: "Meta-Learning for Subject Adaptation in Low-Data Environments for EEG-Based Motor Imagery Brain-Computer Interfaces"
_ICLR.cc/2023/TinyPapers — Submitted to Tiny Papers @ ICLR 2023_

### Official Review · Reviewer_Npbu · 2023-03-24

**Confidence:** 3

**Summary Of Contributions:**

The authors propose a novel method for transfer learning for EEG decoding. In particular, they compare their method (Reptile meta-learning) to a traditional transfer learning approach.

**Rating:**

Great Start (GS): a submission which meets some of the reviewing criteria but has room for improvement

**Strengths And Weaknesses:**

The authors provide an interesting method to approach a well-motivated and important problem in BCI. They clearly explained the problem and why it is a difficult one, as well as some existing approaches. Their methodology is well explained for the most part (see below). They also presented a table of results and explain that their method outperforms the current state of the art for certain numbers of fine-tuning shots.

The main weaknesses of this paper are that some of the writing is unclear, and the specific learning approach could be better explained to make the method more reproducible. Please see the suggested changes section for more specific feedback.

**Suggested Changes:**

Reproducibility:
- I think the methodology section could be built out a bit more. If there is a space issue to fit within two pages, consider getting rid of the section headers for 3.1 and 3.2, since paragraph breaks could also work well. The general three-stage approach and section 3.1 are clear, but I have trouble understanding section 3.2. A short explanation of what the Reptile algorithm is doing (what it takes in, what it modifies, etc) would be helpful here and make the method as a whole more reproducible. It would also be hugely beneficial for the authors to release their code if accepted.

Clarity:
- The abstract currently contains the same first two sentences repeated, once this is remedied it could clear up space for later sections
- The citations are rendering right next to the text which is a bit distracting
- Some sections could use edits for clarity. In particular, I found the end of the related work section to be slightly unclear.
- The results are much appreciated and show the strength of their method in certain contexts. However, visually it could be nice to use something like a bar plot instead, with some way to highlight where their method is doing well. If the authors prefer the table, bolding the number of the better performing method would make reading it easier.

---

> ### Author Response · Authors · 2023-06-01
> **Response to Reviewer Npbu**
>
> We really appreciate your constructive suggestions to improve the paper. We address each of your concerns below.
>
> ### Unclear Writing:
> 1. Thank you for pointing out the repeated section in the abstract. It has been remedied in the revised submission.
> 2. We apologise for the distracting style of citations. We have fixed that in the revised submission.
> 3. We apologise for the unclear related works section. We have provided a comprehensive explanation of the related works and the contribution of our paper, in the revised submission.
> 4. Thank you for appreciating our results table. To better highlight the strength of our method, we have included a table with the average accuracy across all subjects, and bolded the better performing method, in the revised submission. We have also included a graph to visualize the values in the table.
>
> ### Reproducibility:
> 1. We apologise for omitting the details for the specific learning approach. As pointed out, we have removed the extra space from section headers, and explained the detailed algorithm for our meta-learning approach.
> 2. We understand that it would be hugely beneficial if our code is released, which we have linked in the revised submission.
>
> We understand the main weakness of our paper, and we have improved the clarity of writing accordingly and provided detailed explanations along with the code in the revised submission.

---

### Official Review · Reviewer_4ois · 2023-03-26

**Confidence:** 5

**Summary Of Contributions:**

This study proposed using meta-learning algorithm Reptile to initialize the model in an EEG motor imagery classification task. The proposed Reptile-EEGNet method outperforms the EEGNet baseline.

**Rating:**

Needs Clarification (NC): a submission which does not meet the reviewing criteria and needs clarification for its described problem or solution

**Strengths And Weaknesses:**

Strengths

1. The experiment design is valid. The proposed Reptile-EEGNet method outperformed the EEGNet baseline.

Weaknesses

1. Reptile is the key innovation in this work but it is unclear how Reptile works.

2. There is no code provided to replicate the result.

3. The presentation can be improved. (see suggestions below)

**Suggested Changes:**

Suggestions

1. I suggest authors to proofread the manuscript prior to submission. The first paragraph in abstract is redundant. The appendix section is not updated. A URM statement is needed.

2. There are many studies working on EEG motor imagery classification task. The authors should discuss related literature more comprehensively.

3. EEGNet should be cited.

4. It's better to include details of EEG signals, like number of channels, length of time courses.

5. Please consider adding technical description about the Reptile algorithm.

6. Please consider adding quantitative summary in results section.

7. In Table 1, specify accuracy in % and the exact number for "many". It is more visually straight-forward if better performance is highlighted in bold.

---

> ### Author Response · Authors · 2023-06-01
> **Response to Reviewer 4ois**
>
> Thank you for reviewing our paper and your constructive comments. We have updated the revised version accordingly with the following changes:
>
> 1. We have updated the abstract and appendix, removing redundant sections and adding a detailed analysis of the results of the appendix.
> 2. We have added the URM statement.
> 3. We have included a comprehensive coverage of related literature for EEG motor imagery classification.
> 4. The details of the BCI Competition IV 2a dataset, like number of channels, length of trials and sampling frequency, have been included.
> 5. A detailed explanation of our meta-learning algorithm has been included.
> 6. A detailed quantitative analysis of the results has been added to the appendix, and summarised in the results section.
> 7. Thank you for the detailed feedback on improving the table. In the revised version, we have specified that the accuracy is in % and the exact number for "many", along with the motivation behind a comparison between few-shot and many-shot classification. The better performance in the table has been highlighted in bold.

---

### Author Response · Authors · 2023-06-01
**Summary of Paper Revision**

We express our sincere gratitude to all the reviewers for their valuable and constructive feedback, which has served as a source of encouragement and has provided us with insightful suggestions for improvement. The thorough reviews of our paper have greatly contributed to the restructuring process, enhancing its clarity. Additionally, they have enlightened us on effective writing styles and techniques for effectively communicating our work.

According to these suggestions, we have improved the paper and summarized the main changes as follows:
1. Fixed the stylistic issues including the repeated section in the abstract and the citation style.
2. Added a table showing the average performance across the experiments and bolded the better performing method in the tables.
2. Provided a detailed explanation of our Meta Learning algorithm in the Methodology section as well as the Appendix.
3. Updated each section to better frame the problem and motivate our solution in a step-by-step way.
4. Updated the title of the paper and added the URM statement.
5. Added details of the related works and the contribution of our paper.
6. Added graphs supporting our approach, and a detailed analysis of our results to the Appendix and summarised in the Results section.
7. The details of the BCI IV 2a dataset have been included, and the exact number for "many" shots has been specified.
8. Released the code for reproducibility.

We are grateful for the reviewers' invaluable guidance, and we believe that these revisions have significantly strengthened the paper. We hope that our improved work now successfully addresses the concerns raised during the review process.

---

### Meta-Review · Area_Chair_Zv8H · 2023-04-02

**Recommendation:** Invite to archive
**Confidence:** 4

**Metareview:**

The application of a meta-learning algorithm (Reptile) to the problem of decoding imagined movements resulted in improved accuracy on unseen data from new subjects. This is important because of the desirability of the higher accuracies while only using a relatively simple technique.

The reviewers seem to agree that the writing of this paper needs to improve to include more details for clarity and reproducibility. For example, the main algorithm used in the study (Reptile) is not sufficiently introduced or explained. The code doesn't seem to have been made available. There are also stylistic issues and the text needs to be edited and proofread. The authors must be clear about their contribution and that they are applying an existing algorithm (Reptile) to a new problem, while also sufficiently detailing the solution in a clear and step-by-step way, every step of which logically builds on the previous.


**Summary:**

The paper applies an existing meta-learning approach called "Reptile" to improve the cross-subject generalization in a task where imagined movements are decoded from EEG recordings.

**Reason For Not Giving A Higher Recommendation:**

First, I encourage the authors to address the specific points raised by the reviewers. Second, I suggest the authors frame the problem and motivate the solution better. It might be helpful to consider why decoding imagined movements is important, why one would use EEG to achieve this, and why the proposed solution is the natural next step given what's known in the field. The term "motor imagery" hasn't been defined so as to contextualize the problem.

In addition to the reviewers' suggestions on clarity, I also suggest the authors consider removing the word "Reptile" from the title as it can easily mislead an unfamiliar reader into thinking that the paper has something to do with the reptilian brain and how reptiles learn, especially because of the co-occurrence of the term EEG which I assume stands for "electroencephalogram" (which hasn't been, but should be, defined in the text).

**Reason For Not Giving A Lower Recommendation:**

The main thesis of the paper was successful at improving performance in cross-subject variability based on the table presented in the appendix. This makes the paper a great start and promising to improve to a CCR standard once revisions have been made.

---

> ### Author Response · Authors · 2023-06-01
> **Response to Area Chair Zv8H**
>
> Thank you for your review and for acknowledging the potential of the paper. Your valuable comments and suggestions have been helpful in improving the paper. We have revised the paper accordingly. We address your concerns below.
>
> 1. We apologise for the lack of clarity and reproducibility. We have fixed the stylistic issues and provided a detailed explanation of the methods. We have also made the code available, and linked it in the revised version.
> 2. We have updated the Introduction, Methodology and Experiments sections to better frame the problem and motivate each step of our solution.
> 3. We acknowledge that the title of the paper needed to be revised, as it could be misleading. We have updated the revised version with a more appropriate title.

---

### Decision · Program_Chairs · 2023-04-07

Invite to archive